# Cryopreservation Differentially Alters the Proteome of Epididymal and Ejaculated Pig Spermatozoa

**DOI:** 10.3390/ijms20071791

**Published:** 2019-04-11

**Authors:** Cristina Perez-Patiño, Isabel Barranco, Junwei Li, Lorena Padilla, Emilio A. Martinez, Heriberto Rodriguez-Martinez, Jordi Roca, Inmaculada Parrilla

**Affiliations:** 1Department of Medicine and Animal Surgery, Veterinary Science, University of Murcia, 30100 Murcia, Spain; Cristina.perez6@um.es (C.P.-P.); Isabel.barranco@um.es (I.B.); junwei.li@um.es (J.L.); lorenaconcepcion.padilla@um.es (L.P.); emilio@um.es (E.A.M.); parrilla@um.es (I.P.); 2Department of Biology, Faculty of Sciences, University of Girona, 17003 Girona, Spain; 3School of Veterinary Medicine, Yangzhou University, Yangzhou 225009, China; 4Department of Clinical and Experimental Medicine, Faculty of Medicine and Health Sciences, Linköping University, SE-58185 Linköping, Sweden; heriberto.rodriguez-martinez@liu.se

**Keywords:** epididymis, ejaculate, spermatozoa, cryopreservation, proteomics, porcine

## Abstract

Cryopreservation induces differential remodeling of the proteome in mammalian spermatozoa. How these proteome changes relate to the loss of sperm function during cryopreservation remains unsolved. The present study aimed to clarify this issue evaluating differential changes in the proteome of fresh and frozen-thawed pig spermatozoa retrieved from the cauda epididymis and the ejaculate of the same boars, with clear differences in cryotolerance. Spermatozoa were collected from 10 healthy, sexually mature, and fertile boars, and cryopreserved using a standard 0.5 mL-straw protocol. Total and progressive motility, viability, and mitochondria membrane potential were higher and membrane fluidity and reactive oxygen species generation lower in frozen-thawed (FT) epididymal than ejaculated spermatozoa. Quantitative proteomics of fresh and FT spermatozoa were analyzed using a LC-ESI-MS/MS-based Sequential Window Acquisition of All Theoretical Spectra approach. Cryopreservation quantitatively altered more proteins in ejaculated than cauda epididymal spermatozoa. Differential protein–protein networks highlighted a set of proteins quantitatively altered in ejaculated spermatozoa, directly involved in mitochondrial functionality which would explain why ejaculated spermatozoa deteriorate during cryopreservation.

## 1. Introduction

Freezing is the best procedure for long-term sperm preservation. Use of frozen-thawed (FT) spermatozoa is currently the best way to (i) overcome some infertility issues in men [1,2], (ii) accelerate genetic progress in livestock species [3], and (iii) contribute to the survival of endangered species [4]. Despite improvements made in cryopreservation procedures and its inherent value, FT-semen is yet not routinely used neither for most livestock species [5,6] nor humans [7,8]. The main drawback of FT is the lower fertility rates after artificial insemination (AI), compared to natural mating or AI with unfrozen semen [9,10]. Therefore, improving both sperm cryosurvival and fertility outcomes for FT-semen in most mammal species remains a major challenge [11].

Cryopreservation involves sudden osmotic and temperature changes that induce molecular disarrangements in both sperm lipids and proteins [11,12]. A thorough understanding of these sperm molecular disarrangements during the process of cryopreservation is a necessary step to optimize current freezing protocols [13,14]. Regarding proteins, earlier studies evidenced that cryopreservation altered the profile of specific proteins involved in key sperm functions, such as motility, capacitation, oocyte binding ability, and the acrosome reaction [15,16,17,18,19]. These findings encouraged further studies focused on revealing the totality of sperm proteins affected by cryopreservation. The advent of quantitative proteomics has allowed the identification of sets of sperm proteins quantitatively affected by cryopreservation in several mammalian species, including human [20,21], ovine [22,23], and porcine [24]. Though a question remains: how can these sperm proteome changes during cryopreservation explain the loss of functional performance of FT spermatozoa? The present study attempts to clarify this issue using pig spermatozoa as cell model.

In addition to their worldwide relevance for human food industry, pigs are a valuable animal model for biomedical research [25] and are particularly interesting for research involving spermatozoa [26]. Boar sires housed in AI centers are subjected to regular programs of ejaculate collection, delivering copious ejaculations containing large numbers of viable, highly functional spermatozoa for AI. The sires are slaughtered when still healthy and fertile, owing to genetic replacement reasons, thus making possible the collection of cauda epididymal mature spermatozoa at the abbatoir. Interestingly, cauda epididymal mature spermatozoa cryopreserve better than ejaculated pig spermatozoa [27] and their cryosurvival rates are more consistent among boars than those of ejaculated spermatozoa [27,28]; whereas substantial differences are registered among AI boars [29,30,31]. Quantitative differences in some sperm proteins have been directly related to the variability of ejaculate spermatozoa among AI boars to overcome cryopreservation [32,33,34,35].

With this background in mind, the first objective of this study was to evaluate the changes sustained in the proteome of cauda epididymal and ejaculated pig spermatozoa during cryopreservation, comparing unfrozen and FT spermatozoa, where absence/presence of seminal plasma (SP) contact is the major difference. Assuming such changes occur, the second objective was to determine how they would explain differences in cryotolerance between these sperm sources.

## 2. Results

### 2.1. Fresh and Post-Thaw Sperm Parameters

Fresh semen samples met the requirements for AI programs, and quality sperm parameters did not statistically differ between ejaculated and epididymal samples (Appendix A). Table 1 summarizes sperm parameters measured at 30 min post-thawing in the cauda epididymal and ejaculate samples. Percentage of motile (total and progressive) and viable spermatozoa were higher (*p* ≤ 0.001) in epididymal than ejaculated samples. Epididymal samples also showed higher (*p* ≤ 0.001) percentage of viable spermatozoa with high mitochondria membrane potential. In contrast, ejaculated samples showed higher (*p* ≤ 0.001) percentage of viable spermatozoa with high plasma membrane fluidity. The intracellular generation of reactive oxygen species (ROS) was highest (*p* ≤ 0.001) in ejaculated viable spermatozoa.

### 2.2. Protein Repertory

The raw protein dataset of pig spermatozoa used in the present study was generated from a pool of 15 samples of mature spermatozoa derived from cauda epididymis or the ejaculate. A total of 1210 proteins with a cutoff of unused prot score > 1.3 (corresponding to a confidence limit of 95% and a false discovery rate (FDR) < 1%) were identified. The dataset was deposited to the ProteomeXchange Consortium (http://www.proteomexchange.org/) via the PRIDE partner repository with the identifier PXD012984.

The sequential window acquisition of all theoretical spectra (SWATH) approach allowed the quantification of 1065 proteins, all of them present in fresh and FT spermatozoa of both cauda epididymis and ejaculated samples. The relative abundance of the quantified proteins in each sperm source is shown in Appendix A. The two-dimensional principal component analysis (PCA), used to evaluate whether the abundance of the quantified proteins differed among sperm sources, explained 55.6% of the total variance. PC1, explaining 32.6% of the variance, discriminated between epididymal and ejaculated sperm samples, regardless of whether they were fresh or FT. PC2, explaining 20.1% of the variance, discriminated between fresh and FT spermatozoa, regardless of whether they were from epididymis or ejaculate (Figure 1). The heat maps (Appendix A) showed that the three technical replicates of each sperm source formed clearly separated own clusters. The dendrograms of the heat maps also showed that the technical replicates of each sperm source merged into a close cluster, reporting the robustness of the proteomics analysis carried out.

### 2.3. Differentially Abundant Proteins

A total of 140 proteins showed quantitative differences among the sperm sources, 55 of them showing post-translational modifications (PTM). The inventory of these differentially abundant proteins, including organism, encoding genome, and relative amount quantified per sperm source and of the PTM, is shown in Appendix A. The Venn diagram in Figure 2 shows how these quantitative different proteins were distributed among the sperm sources. A total of 44 differentially abundant proteins were overlapped among the four sperm sources. Looking at the nonoverlapped differentially abundant proteins, it was noticeable that the changes in abundance shown by 41 proteins were uniquely related to the transit of the spermatozoa from the cauda epididymis to the ejaculate (Table 2). Thirty-one of these 41 proteins were more abundant in fresh cauda epididymal spermatozoa; the other 10 were more abundant in fresh ejaculated spermatozoa. Focusing on the influence of the cryopreservation process, 32 sperm proteins experienced abundance change uniquely related to cryopreservation (Table 3). Three of these 32 proteins were less abundant in FT spermatozoa from both epididymal and ejaculated samples, compared to fresh samples. Another nine proteins changed uniquely between fresh and FT cauda epididymal spermatozoa, all of them being more abundant in fresh spermatozoa. The remaining 20 proteins showed abundance changes between fresh and FT but only for ejaculated spermatozoa. Ten of these 20 proteins were more abundant in the fresh spermatozoa whereas the remaining 10 were more abundant in the FT spermatozoa. Another 23 proteins showed differences in abundance only between FT spermatozoa of the cauda epididymis and those of the ejaculate (Table 4). Specifically, six of these proteins were more abundant in the FT spermatozoa from the cauda epididymis, and the other 17 in those from the ejaculate.

### 2.4. Bioinformatics

#### 2.4.1. Gene Ontology Enrichment Analysis

The differentially abundant proteins among the four pairwise sperm comparisons did not show differences in the distribution pattern among the functional Gene Ontology (GO) categories specifically related to sperm and reproductive functions included in UniProt KB and DAVID databases (Figure 3). The GO categories with the largest protein number were redox homeostasis and stress response, immune response, and energy/carbohydrate metabolism. 

#### 2.4.2. Protein–Protein Interaction Networks

The Cytoscape software with ClueGO plugin were used to examine the interaction networks of differentially abundant proteins between fresh and FT spermatozoa of both epididymal and ejaculate sources. Protein databases of *Homo sapiens* and *Sus scrofa* were separately explored, allowing functional analysis of the 90% and 70% of the differentially abundant proteins, respectively. The generated networks did not include any differentially abundant proteins between fresh and FT epididymal spermatozoa. In contrast, they included five proteins of those found as differentially abundant between fresh and FT ejaculated spermatozoa, four of them coincident between *Homo sapiens* and *Sus scrofa*. Based on the degree of connectivity, these five proteins, so-called DJ-1, cytochrome C oxidase subunit 2 (Cox2), NADH:ubiquinone oxidoreductase core subunit S2 (Ndufs2), Ubiquinol-cytochrome c reductase complex O (Uqcr10), and mitochondrial 2-oxoglutarate dehydrogenase (Ogdh), were involved in pathways related to mitochondrial activity, redox reaction, energy generation and cellular respiration (Figure 4). DJ-1 was the only of the five proteins exhibiting higher abundance in fresh than in FT spermatozoa. The other four proteins were more abundant in FT spermatozoa.

## 3. Discussion

This study is the first one in mammals comparing the proteome of fresh and FT spermatozoa derived from cauda epididymis and ejaculate of the same sire. The results showed that cryopreservation causes quantitative changes in the boar sperm proteome and that many of the altered proteins differ between both cauda epididymis and ejaculated spermatozoa.

### 3.1. Post-Thaw Sperm Quality and Functionality

Data of post-thaw sperm quality and functionality agree with previous studies, showing that mature boar spermatozoa collected from the cauda epididymis cryosurvives better than those from the ejaculate [27,28]. In addition to lower post-thaw recovery rates of motile and viable spermatozoa in the ejaculate samples, the results also showed that the ejaculate cryosurvival sperm population was less functional than that from the cauda epididymis; which would be linked to increased sensitivity of ejaculated spermatozoa to cryostress [11]. These results clearly indicated that cryopreservation affects distinctly epididymal and ejaculated pig spermatozoa, attributed to the protective effect of epididymal fluid to epididymal spermatozoa and the negative subsequent interaction of these spermatozoa with seminal plasma (SP) during ejaculation, an interaction that would impair the freezability of ejaculated spermatozoa. These putative opposite effects of epididymal fluid and SP on sperm freezability would be linked to quantitative differences in protein composition between both fluids [36]. The proteomic results of this study would support this hypothesis, as we argument for next.

### 3.2. Quantitative Changes in Sperm Proteins

The first proteomics finding of the present study confirmed previously reported proteome quantitative differences between fresh pig spermatozoa from cauda epididymal and ejaculate [35], as 41 sperm proteins showed quantitative differences uniquely related to the transit of mature spermatozoa from the cauda epididymis to the ejaculate. Studies comparing features of epididymal and ejaculated fresh spermatozoa are uncommon, maybe due to the difficulty of collecting mature spermatozoa from both sources from a reproductive healthy male. To the best of our knowledge, only two studies, one in ovine [37] and our own above mentioned in pig [35], compared the proteome of cauda epididymal and ejaculated fresh spermatozoa. In agreement, the present study corroborated the absence of qualitative proteome differences between the spermatozoa of both sources. Moreover, it clearly showed that the documented quantitative differences in the sperm proteome relate to the interaction of spermatozoa with the secretions of the male reproductive accessory glands building the SP during ejaculation. These secretions would play a double and opposite role on the ejaculated spermatozoa. On one hand, they would sequentially wash the sperm surface facilitating the detachment of epididymal adsorbed proteins during sperm maturation and cauda storage and, on the other hand, they would transfer some of its components to the spermatozoa, including SP proteins. Some of the proteins that increased in abundance in fresh ejaculated spermatozoa were identified in the pig SP [38]. Similarly, Pini et al. [37] also showed that the increase in the amount of some proteins in ejaculated ovine spermatozoa with respect to epididymal spermatozoa would be due to exposure of epididymal spermatozoa to SP during ejaculation. Remarkable is the finding that many of the proteins whose abundance shifted between epididymal and ejaculated fresh spermatozoa were involved in redox homeostasis and stress response, structural activity, energy and carbohydrate metabolisms, immune response, and calcium metabolism, which would indicate that their change in abundance could influence sperm performance as previously demonstrated by Perez-Patiño et al. [35]. 

The second proteomics finding of the present study was that cryopreservation remodelled the proteome of pig spermatozoa. This finding was previously evidenced in ejaculated spermatozoa from several mammalian species, as bovine [39], human [20,21], ovine [22,23], porcine [24], and also in bovine epididymal spermatozoa [40]. All these studies found quantitative changes in a significant number of proteins between fresh and FT spermatozoa, although few of these differentially abundant proteins were common across studies [11]. Focusing in pig, Chen et al. [24] identified 41 differentially abundant proteins between fresh and FT spermatozoa, 35 of them showing greater quantity in FT spermatozoa. To clarify the reasons for these quantitative changes in a relatively large number of sperm proteins during cryopreservation remains a challenge. Mature spermatozoa are both transcriptionally and translationally silent [41]. Therefore, modifications in the sperm proteome are mainly quantitative and probably, depending on the interchange of sperm proteins, either adsorbed or structural. These quantitative modifications are normally related to PTMs or to the interaction of spermatozoa with the surrounding environment [42,43]. Although many of the differentially abundant proteins between fresh and FT spermatozoa found in the present study can undergo PTMs, the relevance of such modifications for explaining the results of the present study should be minimal, since PTMs have been contemplated in the proteomic analysis carried out. Regarding interaction with the surrounding environment, cauda epididymal and ejaculated fresh spermatozoa remained stored overnight with its own fluid (i.e., cauda epididymal fluid and SP, respectively) before freezing, facilitating sperm–fluid interaction. In addition, many spermatozoa had undergone membrane damage during cryopreservation, as evidenced by propidium iodide (PI) labeling, a DNA fluorescent label that gets into cells with compromised membrane integrity. This loss of membrane integrity could promote efflux of intracellular components, such as cytoplasmic proteins, which could explain the decrease in abundance experienced by some sperm proteins during the cryopreservation process [21]. The increase in abundance experienced by some sperm proteins during cryopreservation is more difficult to explain. Excluding the existing PTMs, because they were considered in the proteomic analysis carried out in the present study; the other possible hypothesized causes would either be protein degradation or the secondary or tertiary structure changes experienced by some proteins during cryopreservation [21]. In this sense, the efflux of cellular components experienced by the relative high number of spermatozoa dying during cryopreservation could lead to the degradation of proteins of living spermatozoa. Gurupriya et al. [44] showed that cryopreservation changed the activity of proteases and protease-inhibitors of buffalo and bovine semen. Another deleterious cause would be the interaction of spermatozoa with the cryoprotectant. Yoon et al. [40] showed that the usual prefreezing incubation of spermatozoa with extenders containing glycerol induced changes in the proteome of bull epididymal spermatozoa. The influence of the cryoprotectant on the sperm proteome was also demonstrated in human spermatozoa by Bogle et al. [21]. These results and those of the present study clearly highlighted the relationship between the alterations in the sperm proteome and the loss of sperm quality and functionality during cryopreservation. Some of the above studies emphasized that sperm proteins quantitatively altered during cryopreservation could be potential biomarkers of sperm cryo-stress [22,40]. However, the above-cited studies missed to look for direct links between altered proteins and functional changes experienced by spermatozoa during cryopreservation. Regarding distribution into GO categories, the majority of the sperm proteins that changed in abundance during cryopreservation in the above studies had both intracellular and extracellular localization, with a significant percentage of them being membrane proteins. Regarding function, they were involved in a wide range of functions directly or indirectly related to sperm performance, which agree with the results achieved in our study (data no shown). 

The third and most novel proteomics finding of the present study was the demonstration that the quantitative changes in the proteome caused by cryopreservation differed between cauda epididymal and ejaculated spermatozoa. Certainly, only three out of the 32 proteins that changed in abundance between fresh and FT spermatozoa were common for epididymal and ejaculated spermatozoa. Of the remaining 29, the abundance of nine proteins changed between fresh and FT epididymal spermatozoa and the other 20 between fresh and FT ejaculated spermatozoa. This peculiar distribution of the altered proteins would indicate that cryopreservation affected differently epididymal and ejaculated porcine spermatozoa, the latter being the most affected. This fact could explain that quality and functionality post-thawing were worse in spermatozoa from the ejaculate than in those from the cauda epididymis. To test this hypothesis, we first identified the functional allocation of the differentially abundant protein into GO categories directly or indirectly related to sperm and reproductive function. The distribution was relatively similar between cauda epididymal and ejaculated spermatozoa, making it difficult to find any sound relationship between the differentially altered proteins during cryopreservation and the differences in post-thaw sperm quality and functionality between the sperm sources. Therefore, the use of more stringent filters allowed us to test how the differentially abundant proteins between fresh and FT spermatozoa of either source (cauda epididymal or ejaculate) were grouped into known protein–protein network topologies for *Homo sapiens* (largest) and *Sus scrofa* (still limited). The results were striking because the generated networks only included proteins differentially abundant between fresh and FT ejaculated spermatozoa, those most affected by cryopreservation. The network highlighted five of the differentially abundant proteins between fresh and FT spermatozoa from the ejaculate, all of them related to mitochondrial functionality, either involved directly in energy requirements or oxidoreductase activity. Three of them, Ndufs2, Uqcr10, and Cox2, are key mitochondrial electron transport complex proteins specifically involved in complexes I, III, and IV, respectively [45]. DJ-1—a protein deglycase—contributes in the regulation of mitochondrial complex I and it would be involved in the control of oxidative stress [46,47]. Finally, Ogdh is an enzyme of the Krebs cycle that would also be involved in mitochondrial functionality, specifically in ROS generation [48]. Hence changes in the abundance of the above proteins would have a direct influence on sperm mitochondrial functionality, including ATP synthesis and ROS generation. Therefore, the altered abundance of these proteins among ejaculated spermatozoa during cryopreservation could explain the lower mitochondrial membrane potential and the higher ROS affecting motility in ejaculated compared to epididymal FT spermatozoa. 

To conclude, this study showed that the quantitative changes in the proteome experienced by porcine mature spermatozoa during cryopreservation differs between those retrieved from the cauda epididymis and those from the ejaculate from the same individuals; those ejaculated showing a greater number of affected proteins. Among these affected proteins, those involved in mitochondrial electron transport complex would explain why FT spermatozoa derived from the ejaculate produced more ROS and had worse mitochondrial membrane potential and motility than FT spermatozoa from the cauda epididymis. In addition to improve our current knowledge of molecular disarrangements experienced by spermatozoa during cryopreservation, the results of this study can be of value to improve the efficiency of cryopreservation extenders for pig spermatozoa by inclusion of mitochondrial protective agents in their composition.

## 4. Materials and Methods 

The procedures implying boar handling and collection of sperm samples were conducted following international guidelines (Directive 2010/63/EU) and previously approved by the Bioethics Committee of the University of Murcia (research code: 639/2012). Chemicals used for preparing sperm extenders, and fluorescent probes for checking sperm quality and functionality were from Sigma Aldrich Co. (St. Louis, MO, USA).

### 4.1. Boars, Epididymis, and Ejaculates

Ejaculated and epididymal mature sperm samples were collected from 10 healthy, sexually mature (12 to 24 months old) and fertile boars housed under controlled environmental conditions (15–25 °C and 16 h of light) in a Spanish commercial AI center (Topigs Norsvin España, Madrid, Spain). Entire ejaculates (a total of 30, 3 per boar) were collected using the semiautomatic collection procedure Collectis^®^ (IMV Technologies, L’Aigle, France). The boars were slaughtered when still healthy and fertile (Slaughterhouse of Mercamurcia, Murcia, Spain) for reasons of genetic replacement. Their scrotal contents were collected immediately after slaughter and kept in isolated containers (5 °C) until the cauda epididymal content was retrieved at the laboratory through retrograde infusion of air in the ductus deferens [27,35]. All sperm samples, either from the cauda epididymis or the ejaculate, were assessed for morphology and total and progressive motility by using procedures listed below. 

### 4.2. Experimental Workflow

The samples from the cauda epididymis (10 samples, one per boar) and ejaculate (30 samples, 3 per boar) were split in two aliquots immediately after collection. The first aliquot, used as fresh sperm source, was extended (1:3, *v*/*v*) in phosphate-buffered saline (PBS) and centrifuged twice (1500× *g* at room temperature (RT) during 10 min; Megafuge 1.0 R, Heraeus, Hanau, Germany). The resulting sperm pellets were extended in PBS (1000 × 10^6^ sperm/mL), dispensed in cryotubes (2 mL Cryogenic vial, Fisher Scientific, Madrid, Spain) and stored at −80 °C (Ultra Low Freezer; Haier, Schomberg, Ontario, Canada). The second aliquot, used as FT sperm source, was extended (1:1, *v*/*v*) in Beltsville Thawing Solution (BTS, 205 mM glucose, 20.4 mM sodium citrate, 10.0 mM KCl, 15.0 mM NaHCO_3_, and 3.6 mM EDTA), stored at 17 °C overnight and thereafter frozen using the below cryopreservation protocol. Once thawed, the content of three straws for each of the 30 FT sperm samples was pooled and centrifuged at 600× *g* for 20 min at RT on a Percoll monolayer gradient (45% in PBS, *v*/*v*) to separate the spermatozoa from other putative cells, debris, and egg yolk remnants. The resulting sperm pellets were extended in PBS and stored at −80 °C following the same protocol as used for fresh sperm samples. To carry out the proteomics analysis, the sperm samples were thawed at RT, and each of the 4 sperm sources (fresh cauda epididymal spermatozoa, fresh ejaculate spermatozoa, FT epididymal spermatozoa, and FT ejaculated spermatozoa) were pooled generating four single sperm samples, one for each sperm source. Each of the 4 sperm pools was split into three aliquots generating three technical replicates for sperm source. Thereby, proteomics was analyzed in a total of 12 samples.

### 4.3. Sperm Cryopreservation

The sperm samples were centrifuged at 17 °C for 3 min at 2400× *g*, and the resulting sperm pellets were frozen using a standard 0.5 mL-straw freezing protocol described by Alkmin et al. [27]. Briefly, sperm pellets were extended to 1.5 × 10^9^ sperm/mL in a Tris-citric acid-glucose extender (111 mM Trizma Base, 31.4 mM monohydrate citric acid, 185 mM glucose) supplemented with egg yolk (80%/20%, *v*/*v*). Then, cooled to 5 °C for 150 min and re-extended to 1.0 × 109 sperm/mL with the same extender (89.5%, *v*/*v*) plus glycerol (9%, *v*/*v*) and Equex STM (1.5%, *v*/*v*, Nova Chemical Sales, Scituate, MA, USA). The extended sperm samples were packed (0.5 mL polyvinyl chloride French straws, Minitüb, Tiefenbach, Germany) and frozen at −40 °C/min using a controlled rate freezer (IceCube 1810, Minitüb, Germany). The straws, stored in liquid nitrogen tanks (GT40, Air Liquide, Paris, France) for at least two weeks, were thawed in a circulating water bath at 37 °C for 20 s.

### 4.4. Quality and Sperm Functionality Assessments

Fresh sperm samples were evaluated within 2 h after collection and FT sperm samples at 30 min after thawing. Sperm quality was evaluated in terms of morphology (only fresh sperm samples), total and progressive motility and viability. The functionality of viable FT spermatozoa was evaluated in terms of intracellular generation of ROS, fluidity of the plasma membrane and the potential of mitochondrial membrane.

Sperm morphology was evaluated under a phase contrast microscope from sperm samples fixed in buffered 2% formaldehyde solution and recorded as the proportion of spermatozoa with normal morphology. Sperm motility was objectively evaluated using a computer-assisted analysis system (ISASV1^®^, Proiser R + D, Valencia, Spain) operating up to 100 video frames per second. The sperm samples (5 µL containing 100,000 spermatozoa) were loaded on Makler counting chambers (Sefi Medical Instruments, Haifa, Israel) and a minimum of 400 spermatozoa per sperm sample were analyzed. Sperm motility was recorded as the percentages of total motile spermatozoa (average path velocity ≥ 20 µm s^−1^) and the proportion of motile spermatozoa showing rapid and progressive movement (straightness of the average path ≥ 40%). 

Viability and functionality of viable FT spermatozoa were cytometrically evaluated (BD FACSCanto II flow cytometer, Becton, Dickinson, CA, USA) using the settings, software, and procedures described by Alkmin et al. [49]. The cytometry workflow used is shown in Appendix A. Sperm viability, in terms of both plasma and acrosome membrane integrity, was evaluated in sperm samples (100 µL aliquots containing 2 × 10^6^ sperm) incubated (37 °C for 10 min in the dark) with 3 µL of Hoechst 33342 (H-42, 0.05 mg/mL in PBS), 2 µL of PI (0.5 mg/mL stock solution in PBS), and 2 µL of fluorescein-conjugated peanut agglutinin (PNA-FITC, 100 µg/mL stock solution in PBS). Spermatozoa positive to H-42 and negative to PI and PNA-FITC were recorded as viable. The intracellular generation of ROS was measured in terms of hydrogen peroxide (H_2_O_2_) generation in FT-sperm samples (1.5 × 10^6^ sperm in 1 mL of PBS) incubated (38 °C for 30 min in the dark) with 1.5 µL of H-42 (0.05 mg/mL in PBS), 1 µL of 2′,7′-dichlorodihydrofluorescein diacetate (H_2_DCFDA, 1 mM in DMSO), and 1 µL of PI (1 mg/mL in PBS). The mean fluorescence intensity of dichlorofluorescein (DCF; H_2_DCF is oxidized by H_2_O_2_ into DCF) emitted by viable spermatozoa (PI negative) was recorded in the results as fluorescence units (FU) per 10^6^ viable spermatozoa. The fluidity of the plasma membrane was evaluated in FT-sperm samples (1.5 × 10^6^ sperm in 1 mL of PBS) incubated (38 °C for 10 min in the dark) with 1.5 µL of H-42 (0.05 mg/mL in PBS), 1 µL of Yo-Pro-1 (25 µM in DMSO), and 2.6 µL of Merocyanine 540 (M-540, 1 mM in DMSO). Spermatozoa positive to H-42 and M-540 and negative to Yo-Pro-1 were recorded as viable and with high plasma membrane fluidity. Mitochondria membrane potential was evaluated in FT-sperm samples (100 µL containing 3 × 10^6^ sperm) incubated (38 °C for 15 min in the dark) with 3 µL of H-42 (0.05 mg/mL in PBS), 2 µL of PI (0.5 mg/mL in PBS), and 0.5 µL of Mitotracker Deep Red 633 (Mitotracker, 20 µM in PBS of a stock solution of 1 mM in DMSO). The spermatozoa positive to H-42 and Mitotracker and negative to PI were recorded as viable with high mitochondria membrane potential.

### 4.5. Sperm Proteomics

The proteomics analyses were carried out in the Proteomics Unit of the University of Valencia (Valencia, Spain), a member of the PRB2-ISCIII ProteoRed Proteomics Platform.

#### 4.5.1. Protein Extraction 

Protein extraction of sperm samples was conducted as described previously by Perez-Patiño et al. [50]. Briefly, sperm samples were centrifuged (14,000× *g* for 10 min; Eppendorf 5424R, Eppendorf AG, Hamburg, Germany) and the total protein of the resulting sperm pellets were extracted using UTC buffer (7 M Urea, 2 M thiourea, 4% 3-((3-cholamidopropyl) dimethylammonio)-1-propanesulfonate (CHAPS)), supplemented with protease inhibitor cocktail (1%, *v*/*v*). The RC_DC Lowry (Bio-Rad, Richmond, CA, USA) assay was used to determine the concentration of extracted protein. 

#### 4.5.2. SDS-PAGE and In-Gel Digestion

Thirty micrograms of protein per sample was mixed (4:1) with 4× Laemmli sample buffer (Bio-Rad, Hercules, CA, USA) were loaded on one dimensional sodium dodecyl sulfate-polyacrylamide gel electrophoresis (1D SDS-PAGE) to remove the remaining UTC and other interferences before tandem mass spectrometry (MS/MS) analysis. The proteins contained in the 1D SDS-PAGE were in-gel digested following the protocol used by Shevchenko et al. [51], which included (i) trypsin digestion stopped with 10% trifluoroacetic (TFA), (ii) peptide extraction by incubation in pure acetonitrile (ACN) at 37 °C in a shaker for 15 min, and (iii) suspension of resulting peptide mixture in of 2% ACN and 0.1% TFA.

#### 4.5.3. Sequential Window Acquisition of All Theoretical Spectra (SWATH) Analysis

A SWATH analysis was performed by liquid chromatography (LC) using a NanoLC Ultra 1D plus Eksigent (Eksigent Technologies, Dublin, CA, USA) connected to an AB SCIEX TripleTOF 5600 mass spectrometer (AB SCIEX, Framingham, MA, USA) in direct injection mode following the procedure described by Perez-Patiño et al. [50]. Firstly, for the spectral library acquisition, 5 µL of a mixture of the twelve digested samples (2 µL per sample) was loaded on a trap NanoLC precolumn (3-μm particle size C18-CL, 350 µm diameter × 0.5 mm long; Eksigent Technologies, Dublin, CA, USA) and desalted with 0.1% TFA at 3 µL/min for 5 min. Then, the peptides were separated using an analytical LC column (3-μm particle size C18-CL, 75 µm diameter × 12 cm long, Nikkyo Technos Co^®^, Tokyo, Japan) equilibrated in 5% ACN and 0.1% formic acid (FA) (Fisher Scientific). Thereafter, a linear gradient from 5% to 35% of ACN containing 0.1% FA at a constant flow rate of 300 nL/min over 180 min was applied to peptide elution. The TripleTOF operated in data-dependent mode with a time-of-flight (TOF) MS scan from 350 to 1250 *m*/*z*, accumulated for 250 ms TOF followed by 150 ms TOF with the same scan range for MS. The 25 most abundant multiply charged (2+, 3+, 4+, or 5+) precursor peptide ions were automatically selected. Ions with 1+ and unassigned charge states were rejected from the MS/MS analysis. The rolling collision energy equations were automatically set by the instrument according to the equation |CE| = (slope)x(*m*/*z*) + (intercept) with Charge = 2; Slope = 0.0575 and Intercept = 9.

Then, for a label-free quantification using SWATH analysis, each sperm sample was individually analyzed configuring the TripleTOF 5600 (AB SCIEX, Framingham, MA, USA) as described by Gillet et al. [52] for SWATH-MS-based experiments. In this way, 5 µL of one of the three technical replicates from each sample was randomly loaded onto a 3 µL C18-CL trap column (dimensions: 75 μm × 15 cm; NanoLC Column, Eksigent Technologies) and flushed for 5 min with 0.1% trifluoroacetic (TFA) at 3 µL/min to remove salts. Peptides were resolved over a 90 min gradient of 5 to 40% acetonitrile (ACN) at a flow rate of 300 nL/min using an analytical 3-µm C18-CL column equilibrated in 5% ACN and 0.1% FA. The analysis of eluted peptides was carried out in the spectrometer nanoESI qQTOF (AB SCIEX TripleTOF 5600) and the TripleTOF (AB SCIEX) operated in SWATH mode, in which a 0.050 s TOF MS scan from 350 to 1250 *m*/*z* was performed, followed by 0.080 s product ion scans from 230 to 1800 *m*/*z* split into 37 overlapping windows (3.05 sec/cycle). The collision energy for each window was calculated for 2+ charged ion at the center of each SWATH block with a collision energy spread of 15 eV. The MS was always operated in high sensitivity mode.

#### 4.5.4. Protein Quantification

The wiff files obtained from SWATH experiment were analyzed by PeakView^®^ (v2.1, AB SCIEX) and peaks from SWATH were extracted with a peptide confidence threshold of 95%. A FDR less than 1% and 6 transitions per peptide were required to quantify one peptide. The extracted ions chromatograms were integrated, and the peak areas were normalized by total sum, and the sum of all areas was equalized for all the samples. 

#### 4.5.5. Gene Ontology and Bioinformatics Analysis

A GO enrichment analysis was performed for the differentially expressed sperm-proteins using the online bioinformatics tool UniProt KB and DAVID. The UniProt KB database (www.uniprot.org), downloaded 23/07/2018, contains 120,243,849 total entries with 48,936 of them encoded in *Sus scrofa* taxonomy. The DAVID Database for Annotation, Visualization and Integrated Discovery (DAVID Bioinformatics Resources 6.8; https://david.ncifcrf.gov/ [53,54]) integrates numerous public sources of protein annotation and, consequently, contains information of more than 1.5 million genes from more than 65,000 species. The PTMs were explored using Unimod website (http://www.unimod.org/modifications_list.php). Peptides showing exclusively artificial modifications introduced by the sample preparation procedure were manually removed from the list. The functional allocation of the differentially abundant proteins was manually performed and restricted to the GO categories directly or indirectly related to sperm and reproductive functions contemplated in UniProt KB and DAVID databases. GO enrichment and network visualization and analysis were performed with Cytoscape v3.2.1 [55] and ClueGO plugin v2.2.3 [56] using the enrichment/depletion multiple two-sided hypergeometric test with Bonferroni step down correction and a Kappa score (term/pathway connectivity) of 0.4. Only enriched functions with a *p*-value < 0.05 were considered. The other ClueGO parameters were kept to default settings.

### 4.6. Statistical Analysis

The influence of sperm source (cauda epididymis vs. ejaculate) in post-thaw sperm quality and functionality was evaluated using a mixed ANOVA test (IBM SPSS v24.0 software, IBM Spain, Madrid) including boar and ejaculate as random effects. In proteomics, the quantitative data processed by PeakView^®^ were further exported to MarkerView^®^ (v1.2, AB SCIEX) for quantitative and statistical analysis. The data across the runs were normalized using total area sum. Then, the Multiexperiment Viewer (MeV) software (version 4.8) (http://www.tm4.org) was used to identify quantitative differently abundant sperm proteins applying a Student’s t-test. Differences in protein abundance were considered with an adjusted *p*-value ≤ 0.01, and only those with a fold change (FC) ≥ ±1.5 after log_2_ transformation were recorded. The explanatory ability of different abundant proteins was illustrated by mean of heat maps after z-score normalization, using Euclidean distances. The online software ‘FunRich’ (http://www.funrichweb.org) was used to show how the differently abundant proteins were distributed among the sperm sources. Differences of differentially abundant proteins into the functional GO categories among sperm sources were analyzed using a chi-squared analysis with an adjusted *p*-value ≤ 0.05.

## Figures and Tables

**Figure 1 ijms-20-01791-f001:**
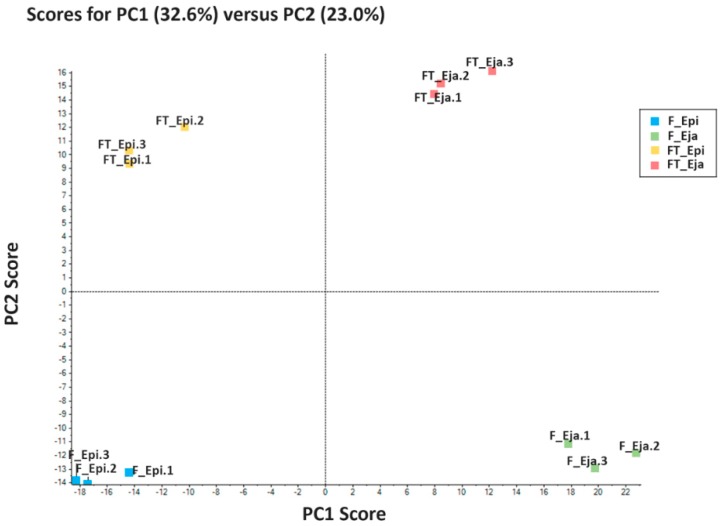
Two-dimensional principal component analysis (PCA) evaluating differences on the quantified proteins among sperm sources. PC1 discriminates between epididymal (Epi) and ejaculated (Eja) spermatozoa, whereas PC2 between fresh (F) and frozen-thawed (FT) spermatozoa. The squares represent the three technical replicates for each sperm source, based on the relative amounts of protein quantified in each one of them.

**Figure 2 ijms-20-01791-f002:**
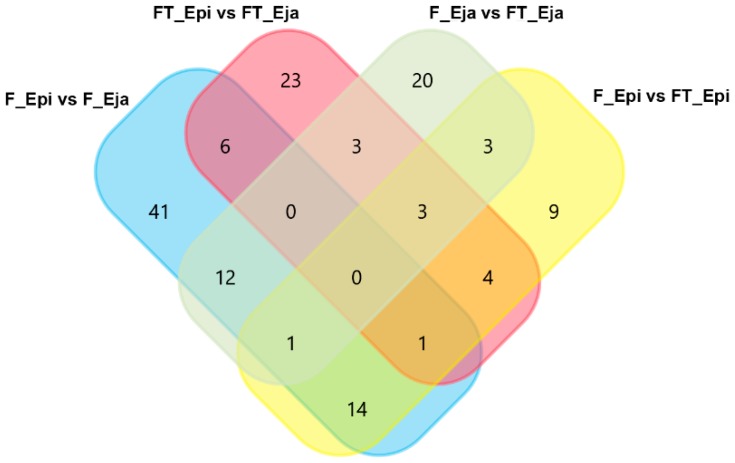
Venn diagram depicting the distribution of differentially abundant proteins among fresh (F) and frozen-thawed (FT) spermatozoa collected from cauda epididymis (Epi) and the ejaculate (Eja).

**Figure 3 ijms-20-01791-f003:**
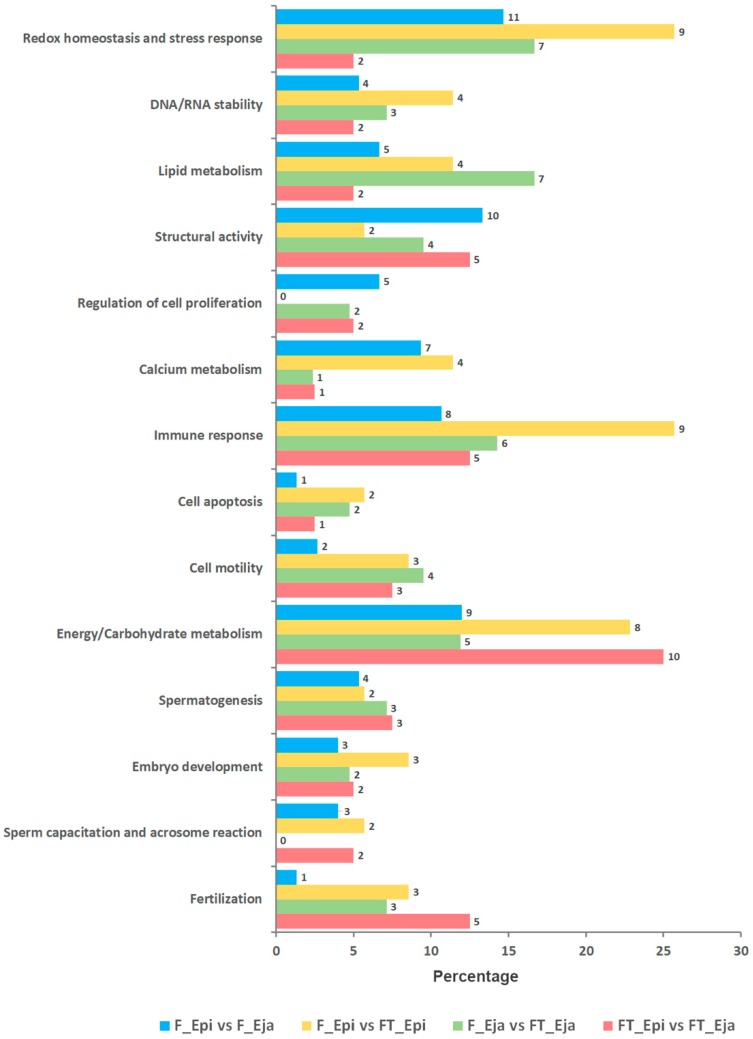
Bar chart depicting the Gene Ontology (GO) distribution of the differentially expressed sperm proteins according to their annotated function in UniProt KB and DAVID databases. Only categories directly or indirectly related to sperm and reproductive functions were considered. Fresh (F) or frozen-thawed (FT) mature boar spermatozoa collected from cauda epididymis (Epi) or ejaculate (Eja). Bars show the percentage of proteins in each functional group and the number in front of each bar indicates the number of proteins.

**Figure 4 ijms-20-01791-f004:**
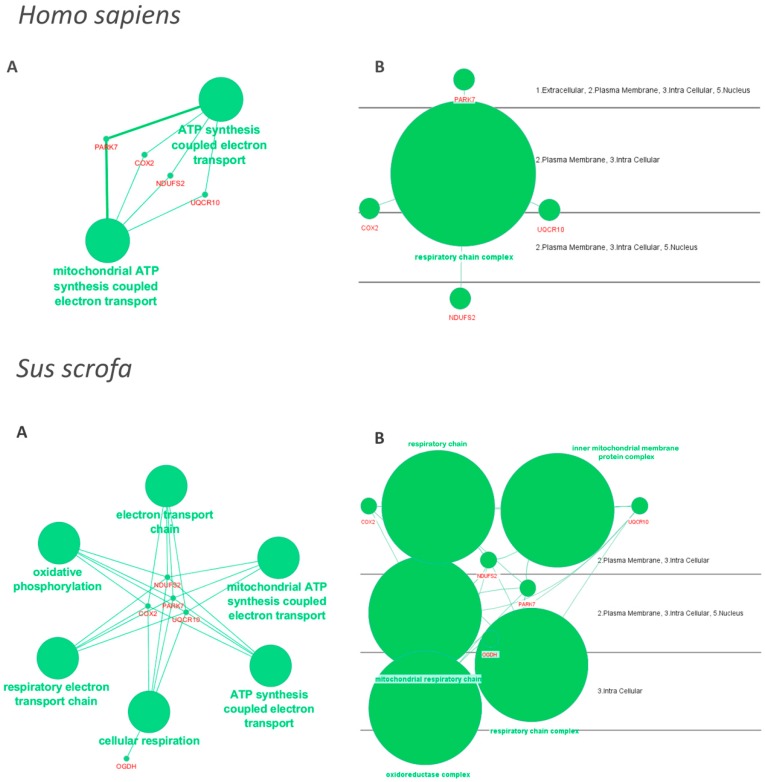
Networks highlighting the most relevant protein–protein interactions among those differentially abundant pig sperm proteins, comparing fresh and frozen-thawed ejaculated spermatozoa. The network was generated using Cytoscape and the ClueGO plug-in considering the database proteins of Homo sapiens or Sus scrofa. The networks were generated for functional (**A**) and subcellular localization (**B**) categories with the terms represented as nodes linked based on their κ-score level (≥0.4) is shown. The node size represents the relevance of enrichment.

**Table 1 ijms-20-01791-t001:** Post-thawing parameters (mean ± SEM) of mature pig spermatozoa collected from the cauda epididymis and ejaculates of 10 boars.

Sperm Parameters ^1^	Sperm Source	
	Cauda Epididymis	Ejaculate
Total motility (%)	60.50 ± 0.63 ^a^	46.83 ± 1.04 ^b^
Progressive motility (%)	43.60 ± 0.70 ^a^	36.23 ± 1.03 ^b^
Viability (%)	62.19 ± 0.85 ^a^	48.40 ± 0.84 ^b^
Viable sperm with high plasma membrane fluidity (%)	3.44 ± 0.33 ^a^	7.83 ± 0.36 ^b^
Viable sperm with high mitochondria membrane potential (%)	52.71 ± 0.88 ^a^	47.99 ± 0.90 ^b^
Intracellular ROS^2^ generation in viable sperm (fluorescent units per 10^6^ viable sperm)	1826.71 ± 102.04 ^a^	2778.35 ± 116.84 ^b^

^1^ Measured at 30 min post-thawing. ^2^ ROS: Reactive oxygen species. ^a,b^
*p* ≤ 0.001.

**Table 2 ijms-20-01791-t002:** Proteins differentially abundant between fresh pig spermatozoa collected from cauda epididymis (Epi) and ejaculate (Eja).

Entry Name	Protein Name	Gene Name	FC* log2 Epi vs. Eja
M3YIY8_MUSPF	Solute carrier family 25 member 3	*SLC25A3*	5.61
Q1RLJ4_PIG	Prepro-beta-defensin 129 protein	*DEFB129*	4.15
F1S086_PIG	Solute carrier family 25 (Aspartate/glutamate carrier), member 12	*SLC25A12*	2.93
A0A286ZQW4_PIG	Nucleoporin 210 like	*NUP210L*	2.83
F1RRA6_PIG	Solute carrier family 25 member 31	*SLC25A31*	2.66
W5P5P6_SHEEP	Calcium regulated heat stable protein 1	*CARHSP1*	2.63
L8ISZ2_9CETA	Uncharacterized protein	*M91_05957*	2.51
G1PY44_MYOLU	Saccharopine dehydrogenase (putative)	*SCCPDH*	2.46
H9H013_HORSE	Solute carrier family 25 member 1	*SLC25A1*	2.34
U3CNV1_CALJA	ADP/ATP translocase 3	*SLC25A6*	2.31
I3LBP3_PIG	Maestro heat like repeat family member 2B	*MROH2B*	2.25
I3LR09_PIG	Cell division cycle 14B	*CDC14B*	2.22
F1RLC5_PIG	Ferritin	*FTMT*	2.18
A0A287BD08_PIG	Uncharacterized protein	*_*	2.06
F1SKK4_PIG	Transmembrane protein 89	*TMEM89*	1.95
F1S8P1_PIG	Uncharacterized protein	*SCCPDH*	1.9
W5PLN5_SHEEP	Uncharacterized protein	*_*	1.9
F1SAS2_PIG	Acyl-CoA synthetase short chain family member 1	*ACSS1*	1.86
F1RX00_PIG	Amine oxidase	*MAOA*	1.84
A0A287AN57_PIG	Lipocalin 12	*LCN12*	1.83
I3LPR1_PIG	Cancer/testis antigen 83	*CT83*	1.81
A0A287ATC1_PIG	Uncharacterized protein	*_*	1.75
A0A287BJG7_PIG	Uncharacterized protein	*_*	1.74
Q2I373_PIG	Fascin	*FSCN3*	1.73
F1RVC1_PIG	Glycerol kinase 2	*GK2*	1.71
H0V2U0_CAVPO	Frataxin	*FXN*	1.66
A0A287A9I6_PIG	SAMM50 sorting and assembly machinery component	*SAMM50*	1.65
A0A287BSC3_PIG	Alkaline phosphatase	*ALPL*	1.62
F1S7L6_PIG	UBX domain protein 6	*UBXN6*	1.62
F1SN95_PIG	Nucleoporin 155	*NUP155*	1.6
M3VZT4_FELCA	Dynein axonemal intermediate chain 1	*DNAI1*	1.57
A0A287D4X0_ICTTR	Dynein light chain roadblock	*DYNLRB2*	−2.02
I6R469_PIG	Calcium binding tyrosine-(Y)-phosphorylation regulated transcript variant 3	*CABYR*	−2.17
F1RG35_PIG	STIP1 homology and U-box containing protein 1	*STUB1*	−2.42
A0A287BNS1_PIG	Chromosome 10 open reading frame 82	*C10orf82*	−2.45
F1S137_PIG	Pro-epidermal growth factor	*EGF*	−3.7
A0A2C9F3H7_PIG	Dipeptidyl peptidase 4	*DPP4*	−3.83
I3LH70_PIG	Collagen type XVIII alpha 1 chain	*COL18A1*	−3.95
Q8WNW8_PIG	Nexin-1	*PN-1*	−6.99
I7HJH6_PIG	Seminal plasma sperm motility inhibitor	*AQN-3 SPMI*	−7.01
A0A286ZY95_PIG	Fibronectin 1	*FN1*	−9.54

* FC: Fold change.

**Table 3 ijms-20-01791-t003:** Proteins differentially abundant between fresh (F) and frozen-thawed (FT) pig spermatozoa from cauda epididymis (Epi) and ejaculate (Eja).

Entry Name	Protein Name	Gene Name	FC* log2	
			F vs. FT_Epi	F vs. FT_Eja
K7GMV8_PIG	Ectonucleotide pyrophosphatase/phosphodiesterase 3	*ENPP3*	3.74	_
M3W2V1_FELCA	Thioredoxin	*TXN*	3.6	_
E1CAJ5_PIG	Protein disulfide-isomerase	*grp-58*	3.15	_
D0G6X8_PIG	Beta-hexosaminidase	*HEXB*	2.9	_
W5P708_SHEEP	Actinin alpha 4	*ACTN4*	2.87	_
G9F6X8_PIG	Protein disulfide-isomerase	*P4HB*	2.74	_
F7HXH1_CALJA	Phosphoglycerate mutase	*PGAM2*	2.27	_
V9HWB4_HUMAN	Epididymis secretory sperm binding protein Li 89n	*HEL-S-89n*	2.23	_
K7GRY0_PIG	Ubiquitin like modifier activating enzyme 1	*UBA1*	2.08	_
M3WST2_FELCA	Glutaminyl-peptide cyclotransferase	*QPCT*	3.26	2.6
A0A287A5M7_PIG	Aldose reductase	*AKR1B1*	1.74	1.97
F1SAT2_PIG	Cystatin	*CST11*	1.52	1.58
MSMB_PIG	Beta-microseminoprotein	*MSMB PSP94*	_	2.85
A0A287AXJ7_PIG	Inositol-3-phosphate synthase 1	*ISYNA1*	_	2.82
F7E460_HORSE	Uncharacterized protein	*RELCH*	_	2.64
A0A287BM88_PIG	Lipocalin 8	*LCN8*	_	2.32
Q307R2_RABIT	Peptidyl-prolyl cis-trans isomerase (Fragment)	*PPIA*	_	2.23
A0A1B2TT55_PIG	Aspartate aminotransferase	*GOT1*	_	1.97
U3F9K8_CALJA	26S proteasome non-ATPase regulatory subunit 1 isoform 1	*PSMD1*	_	1.83
Q0R678_PIG	DJ-1 protein	*PARK7*	_	1.71
Q8WNR3_PIG	Arylsulfatase A	*AS-A*	_	1.52
A0A287ARR1_PIG	Ferritin	*FTH1*	_	1.51
Q2EN79_PIG	Ubiquinol-cytochrome c reductase complex O	*UQCR10*	_	−1.69
F1SDZ2_PIG	Uncharacterized protein	*GALC*	_	−1.77
A0A0K0KW08_PIG	L-lactate dehydrogenase	*LDHAL6B*	_	−1.81
F1S1A8_PIG	NADH:ubiquinone oxidoreductase core subunit S2	*NDUFS2*	_	−1.92
F1RWZ8_PIG	Dual specificity phosphatase 21	*DUSP21*	_	−1.97
I3LNF2_PIG	Dynein axonemal heavy chain 1	*DNAH1*	_	−2.2
F1S0P1_PIG	Regulator of G protein signaling 22	*RGS22*	_	−2.22
K9IVI1_PIG	2-oxoglutarate dehydrogenase, mitochondrial	*OGDH*	_	−2.45
V5KX18_PIG	Cytochrome c oxidase subunit 2	*COX2*	_	−2.51
G3QF37_GORGO	Chromosome 9 open reading frame 84	*C9ORF84*	_	−3.8

* FC: Fold change.

**Table 4 ijms-20-01791-t004:** Proteins differentially abundant between frozen-thawed pig spermatozoa from cauda epididymis (Epi) and ejaculate (Eja).

Entry Name	Protein Name	Gene Name	FC* log2 Epi vs. Eja
A0A287BPZ5_PIG	Uncharacterized protein	*IQCN*	3.08
A0A2I3T063_PANTR	Uncharacterized protein	*_*	2.68
A0A287BCS4_PIG	Pitrilysin metallopeptidase 1	*PITRM1*	2.63
I3LJA4_PIG	Uncharacterized protein	*_*	2.15
A0A286ZN09_PIG	Uncharacterized protein	*_*	1.9
A0A287AI93_PIG	Uncharacterized protein	*IQCN*	1.8
W5Q086_SHEEP	cAMp-dependent protein kinase type I-alpha regulatory subunit	*PRKAR1A*	−1.6
ENPL_PIG	Endoplasmin	*HSP90B1*	−1.61
G1M5S6_AILME	Cytochrome b-c1 complex subunit 6	*LOC100469420*	−1.77
F1S1R1_PIG	Cylicin 1 OS = Sus scrofa	*CYLC1*	−2.22
I3LNH3_PIG	Neutral alpha-glucosidase AB	*GANAB*	−2.28
A7VK02_PIG	Transmembrane protease, serine 2	*TMPRSS2*	−2.32
W5Q770_SHEEP	Uncharacterized protein	*_*	−2.42
W5QAX3_SHEEP	Actin gamma 1	*ACTG1*	−2.74
A0A286XIX4_CAVPO	Fibronectin 1	*FN1*	−3.29
A0A250Y012_CASCN	Glia-derived nexin	*SERPINE2*	−3.3
A0A287B423_PIG	Uncharacterized protein	*SPESP1*	−3.52
M3Z880_MUSPF	Peptidyl-prolyl cis-trans isomerase	*PPIA*	−3.64
Q29057_PIG	Glutathione S-transferase	*GST*	−3.67
F1PLM3_CANLF	Tubulin alpha chain	*TUBA1A*	−4.18
F1RT83_PIG	Uncharacterized protein	*SDCBP*	−4.91
Q4R0H6_PIG	Spermadhesin PSP-I	*PSP-I*	−5.72
A0A0A8IK66_PIG	Heparin-binding protein WGA16	*WGA16*	−6.58

* FC: Fold change.

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
