# Peer review of "Cryopreservation Differentially Alters the Proteome of Epididymal and Ejaculated Pig Spermatozoa"

_ijms, 2019, doi:10.3390/ijms20071791_

Reviewer 1 Report

The authors answered the opponents' comments satisfactorily and improved the manuscript.

Reviewer 2 Report

Dear authors,

thank you for the reply to my comments and suggestion.

All of the comments were updated and discussed.

I recommend publishing this manuscript in this revised form.

Congratulation, Great work!

My best regards.

Reviewer 3 Report

Revised version addresses all the concerns

This manuscript is a resubmission of an earlier submission. The following is a list of the peer review reports and author responses from that submission.

Round  1

Reviewer 1 Report

This work presents the proteomic study on epididymal and ejaculated pig spermatozoa focused on the impact of cryopreservation on sperm protein profile. The results are well discussed.

Comments and suggestions:

Please describe clearly the Fold change (FC), which sperm samples are compared, in the heading of Table 2 and 4, such as in the Table 3 (F vs FT).

I suggest to show in the supplementary pictures a representative sample of the spermatozoa for investigating the F and FT sperm viability and functionality.

The authors should explain the principle of the PM fluidity measurement in the Method part (p. 15, lane 453).

Chapter of the Protein extraction (p. 15) is not appropriate. There are described two different things - protein extraction and SDS electrophoresis followed by MS analysis. Please change the heading of chapter, and better describe and arrange these methods. Additionally, I think that centrifugation of sperm samples has not been too gentle to preserve the integrity of acrosome. Did you check it? It is not clear from methods, although the authors describe the incubation of sperm with PNA lectin. Please explain this methodological intention. Sperm proteins were extracted by urea. The authors described that samples were loaded on the gel electrophoresis directly. Did not use the sample buffer for electrophoresis?

Please add titles into supplementary tables S2 and S3 and describe headings.

Reviewer 2 Report

Dear authors,

 I have read your excellent manuscript " Cryopreservation differently alters proteome in epididymal and ejaculated pig spermatozoa” with great interest.

I have found this research very remarkable, in my opinion having the high potential for the future. Mainly in the field of setting up novel media for cryopreservation of spermatozoa or other cells, as well. 

However, I have found some things that could be corrected or discussed more in detail in your manuscript. I am going to explain my opinions and suggestions/recommendations in common order.

 Abstract

 lns 19-21 -  I recommend to check the sentence and separate it, in my opinion, it is too long. 

 ln 26 - I recommend to explain the SWATH. As a general rule, in Abstract there could be used commonly used abbreviations.

 Introduction

 ln 36 - I recommend to check the use of word aid 

ln 69 - I recommend to authors to move the last part of the sentence "where absence/presence of SP contact is the major difference" to the paragraph above.

 Results

 tab. 1 - According to the general rules for Tables, Figures description I suggest to check the heading of Table. Time of incubation of spermatozoa after thawing is missing.

 Venn diagram - I suggest highlighting the most important results

 Discussion

 lns 250 - 252 - Are there in vivo studies dealing with these differences in sperm functionality (cryosurvival) ?

 lns 304 - 305 - Did authors try shorter or longer incubation? 

 Conclusion - What authors suggest to future experimental work e.g. regarding semen extenders?

 Material and Methods

 ln 399 - I suggest to change abbreviation rt to RT

 ln 416 - 2 4000 x g is really high centrifugation force. This is a mistake, isn`t it?

 ln 440 - for progressive spermatozoa there was only threshold STR 40%?

 ln 446 - do authors have also data of acrosomal integrity?

Reviewer 3 Report

In this study authors describe an interesting comparison among epididymal and ejaculated boar sperm in both fresh (F) or after frozen/thawed (FT) treatment.

Results show how proteins are differentially abundant between epididymal (EPI) compared to ejaculate (EJA) sperm, with particular relevance to the implications that frozen/thawed (FT) treatment introduces to each sample.

Analyzes performed were adequate and carefully carried out as indicated by the close clusters of replicates reported for both each sperm sample and each sperm source.

 Minor revision

·         Table 1 and Table 1 S should be merged and differences between fresh and FT sperm compared, analyzed, and better descripted in the Results.

·         At the light of what evidenced from the comparative analysis about the five proteins differentially abundant between the F and FT ejaculate spermatozoa and involved in mitochondrial function/regulation, a sentence better describing differences should be added in the Results.

·         Some English typewriting errors.

·         Period from lane 256-260 should be better explained.
